# Protein Glycosylation as Biomarkers in Gynecologic Cancers

**DOI:** 10.3390/diagnostics12123177

**Published:** 2022-12-15

**Authors:** Hung Shen, Chia-Yi Lee, Chi-Hau Chen

**Affiliations:** 1Department of Obstetrics and Gynecology, National Taiwan University Hospital, Hsin-Chu Branch, Hsin-Chu City 300, Taiwan; 2Department of Obstetrics and Gynecology, National Taiwan University College of Medicine and Hospital, Taipei 100, Taiwan

**Keywords:** glycosylation, ovarian cancer, endometrial cancer, cervical cancer, biomarker

## Abstract

Gynecologic cancers are the leading cause of death in women. Endometrial, ovarian, and cervical cancer are the three main types of gynecologic cancers. Poor prognoses and high mortality rates of advanced-stage cancer are still challenges of all three types. Diagnostic tools for early cancer detection could be the cornerstone for further cancer treatment and prevention. Glycosylation plays a vital role in cell proliferation, adhesion, motility, and angiogenesis, and is aberrantly expressed in cancer cells. Alterations of glycosylation may represent promising biomarkers with potential diagnostic and monitoring applications, as well as disease prognosis. Many glycosylated biomarkers, including glycoprotein, glycan, and enzyme, were discovered and well-studied for application in gynecologic cancers. Some of them have been developed as targets for cancer treatment. The use of certain biomarkers for diagnostics and monitoring of gynecologic cancers has clinical advantages, as it is quantitative, comparable, convenient, and inexpensive. However, one of the single markers have sufficient sensitivity for the screening of gynecologic cancers. In this review, we introduced the details of glycosylation and the current application of glycosylated biomarkers in these three cancers. Moreover, we also reviewed the different roles of each biomarker in other cancers and aimed to understand these glycosylated biomarkers comprehensively.

## 1. Introduction

When cancer originates in a woman’s reproductive organs, it is called gynecologic cancer. Gynecologic cancers are one of the leading causes of cancer-related deaths in women worldwide. The three main types of gynecologic cancer are endometrial, ovarian, and cervical. The rarer type of gynecologic cancers (vulva and vaginal cancer) will not be covered in this review. Endometrial cancer is the most common gynecologic cancer in developed countries, and its incidence is rising [1,2]. Patients are often diagnosed when the disease is still confined to the uterus. The early diagnosis resulted in a higher survival rate of 95% in localized diseases. However, patients with distant metastasis have a dismal prognosis, with 5-year survival rates of less than 20%. Despite the tendency toward early diagnosis, 13–17% will develop a recurrence resulting in a poor prognosis [3]. Ovarian cancer has the highest mortality rate among gynecologic cancers, with a 5-year survival rate of 27% for stage III and 13% for stage IV ovarian cancer [1,4]. The poor survival rate is due mainly to the lack of a reliable early detection method, tendency to metastasize at an early stage, and a resistance to available therapeutic interventions [2,3]. Cervical cancer is currently the third-most common gynecologic cancer in Taiwan. Cervical cancer has become rare in high-income countries, but it is still the main cancer-related cause of mortality among women in low- and middle-income countries. In 2020, the global estimates of new cases were 0.6 million, with approximately 0.34 million deaths from this neoplasm [1]. Although preventive methods such as cervix cancer screening and the human papillomavirus (HPV) vaccine are now widely used, the morbidity associated with cervical cancer remains relatively high, such as in Sub-Saharan Africa.

Implementing screening and early detection programs is one of the cornerstones of cancer prevention. Non-invasive biomarkers, such as those from serum, could provide a helpful complement to imaging and cytology diagnostic methods, and have the potential to aid clinical decisions as part of a routine blood test. The current clinical biomarker used in managing ovarian cancer is the serum marker cancer antigen 125 (CA125), which, although used widely for disease monitoring, does not provide adequate accuracy for early detection and diagnosis. A recent study has shown that the use of CA125 as a screening marker for ovarian cancer did not improve disease survival but may decrease th incidence of advanced stage ovarian cancer [4,5].

Approximately half of all human proteins are glycosylated, and most U.S. Food and Drug Administration (FDA)-approved cancer biomarkers comprise glycoproteins or carbohydrate antigens [6,7]. Glycosylation serves as a crucial process that influences cell proliferation, adhesion, motility, and angiogenesis. Aberrant glycosylation in cancer cells may affect the progression of cancers. Glycan alterations, such as truncation or modified branching patterns, may correlate with cell growth and enhanced metastasis capacity of cancer cells [7,8]. Thus, glycosylation modification of proteins expressed on the cell surface, or secreted by cancer cells, are promising sources of potential biomarkers.

Recently, glycosylation-based biomarkers in gynecologic cancer research have gained much attention as a tool for cancer prediction and diagnosis based on rapid advances in molecular biology [9]. However, due to the molecular heterogeneity of tumors from patient to patient and the fact that an individual organ can contain tumors of several stages within the same tissue, target proteins and innovative application of glycosylation in gynecologic cancers remain undiscovered [10]. This review will outline our comprehension of protein glycosylation in gynecologic cancers and concentrate on clinically used biomarkers that can serve as prognostic or predictive parameters to assist an individualized treatment approach. Although some cancer biomarkers are elevated in benign diseases and some are undetectable in early-stage cancers, increasing the understanding of existing biomarkers might contribute to the introduction of a combination of different markers for better diagnostic, monitoring, and prognostic performance, as well as the potential discovery of new biomarkers.

## 2. Method

We searched the literature through MEDLINE (PubMed Database) and started in 2013. The last database search was performed on 15 July 2022. We used the combination of the following key words: “ovarian cancer”, “endometrial cancer”, “cervical cancer, “biomarker”, and “glycosylation”, which led to 139 articles. After reviewing the full texts and evaluating the quality of the studies, we finally selected 68 articles. To broaden our search, reference lists of articles already withheld were also verified. Thirty-seven more articles were included through this snowball method. In total, 105 articles were included in this review.

## 3. Definition of Glycosylation

Glycans, also called polysaccharides, are carbohydrate-based polymers made by all living organisms. Glycans can be covalently attached to proteins and lipids to form glycoconjugates. Glycosylation is an enzymatic process that adds sugars to other glycans, proteins, or lipids. It is one of the most common and essential posttranslational modifications on proteins and lipids, and its influence on biological processes is immense. Protein glycosylation includes the addition of *N*-linked glycans, *O*-linked glycans, phosphorylated glycans, glycosaminoglycans, and glycosylphosphatidylinositol (GPI) anchors to peptide backbone [11]. The two most common glycosylations are *N*-linked and *O*-linked (Figure 1).

### 3.1. O-Linked Glycosylation

In *O*-linked glycosylation, glycans attach to the hydroxyl oxygen of serine/threonine (Ser/Thr) residues on target proteins and extend to produce various core and terminal structures that can be sialylated or fucosylated. Mucin-type *O*-glycosylation, consisting of glycans attachment via *O*-linked *N*-acetylgalactosamine (GalNAc) to Ser/Thr residues, is particularly relevant to mucosal sites such as the airways, urogenital and gastrointestinal tracts. These mucosal tissues highly express mucins creating the foundation upon which long and more complex oligosaccharide chains are built. In addition to mucin type *O*-linked GalNAc, Ser/Thr residues may be glycosylated to *O*-fucose, *O*-glucose, *O*-mannose, and *O*-xylose.

Mucins (MUCs) are a family of high molecular weight, heavily *O*-linked glycoconjugates produced by epithelial tissues in most animals [12]. Currently, 22 members of the MUC family have been identified. They can be subclassified as either secreted mucins (MUC 2, 5AC, 5B, 6, 19) or transmembrane mucins (MUC1, 3A, 3B, 4, 12, 13, 15, 16, 17, 18, 20, and 21) depending on their structure and localization [13,14]. In addition to forming an extracellular protective layer over the organs and working as a barrier against external pathogens, mucins have a role in signaling, monitoring, and repairing damaged epithelia [15,16]. Abnormality of mucin expression and structure contributes to biological properties related to human cancer progression. Mucins and mucin-like molecules derived from pathogens are potential diagnostic markers and targets for therapeutic agents.

### 3.2. N-Linked Glycosylation

In *N*-linked glycosylation, glycans are transferred co-translationally to the amide group of asparagine (Asn) residue in the consensus sequence Asn-X-Ser/Thr (X is any amino acid except for Proline). *N*-glycans contain a common pentasaccharide core region consisting of three mannose and two N-acetylglucosamine (GlcNAc) subunits. This can be further modified by adding terminal galactose (galactosylation), GlcNAc (GlcNAclyation), fucose (fucosylation), and sialic acid (sialylation) moieties. *N*-glycans are found in most living organisms and have a crucial role in regulating many intracellular and extracellular functions [11].

## 4. Protein Glycosylation Markers for Gynecologic Cancers

### 4.1. Glycoprotein Markers

Glycoproteins carrying specific glycan structures have been reported in cancer cells. These distinct particles may be secreted or shed into the circulation and work as potential biomarkers. For example, previous research has identified a series of mucins aberrantly secreted by ovarian cancer cells, including MUC1, MUC2, MUC4, MUC13, MUC16, and MUC20 [17,18,19,20]. The shed extracellular domains of some membranous mucins (MUC1 and MUC16) can be detected in serum, and thus may be applicable for cancer diagnosis. Other non-mucin glycoproteins, i.e., alpha-fetoprotein (AFP), carcinoembryonic antigen (CEA), and squamous cell carcinoma antigen (SCC-Ag), are well-documented tumor markers in different kinds of cancer (Figure 2) [21,22,23].

#### 4.1.1. Cancer Antigen 125 (CA125)

CA125, a highly glycosylated MUC16 epitope, was highly upregulated in ovarian cancers. Identified in 1981, it quickly became one of the first serum biomarkers for ovarian cancer [24]. In biochemical structure, CA125 is the largest protein in the family of transmembrane mucin and is a highly *N*- and *O*-glycosylated transmembrane glycoprotein expressed on the epithelium of the Mullerian duct and the lining cell of peritoneum, pleural, or pericardium [25]. The transmembrane proteins have many interactions with other protein partners and regulate different cellular or molecular events [24]. CA125 interacts with mesothelin, facilitating cell—cell adhesion and promoting ovarian cancer metastasis. The interaction between CA125 and mesothelin relies on the *N*-glycosylation of both molecules [26]. CA125 also interacts with galectin-3 and galectin-1 [26,27] and plays a role in cell—cell binding, signal transduction, and protein stabilization [24].

The cut-off of normal values for CA125 was 35 IU/ml, which could include 99% of healthy individuals [28]. Increased CA125 could be found not only in ovarian malignancy but also in benign diseases (e.g., endometriosis), other malignancies, or even physiological conditions [24]. As a result, CA125 has limited sensitivity and specificity as a serum biomarker for ovarian cancer screening [29]. A meta-analysis by Ferraro et al. found an overall sensitivity of 79% and specificity of 78% [30]. On the other hand, CA125 was a good tool for monitoring therapeutic effects and prognosis. The current consensus is to measure the CA125 level for monitoring therapeutic response and recurrence of disease [31].

Moreover, CA125 had been a treatment target for ovarian cancer. Abagovomab is a murine monoclonal anti-idiotypic antibody (molecular weight: 165–175 kDa) that functionally imitates the tumor-associated antigen, CA125 [32]. Although this antibody successfully induced the host’s immune response to CA125, the final phase III trial did not find a survival benefit compared with placebo as a maintenance treatment for advanced ovarian cancer [33].

CA125 is also expressed by other malignancies, especially other gynecologic cancers such as endometrial and cervical. Currently, there is no reliable screening tool or biomarker for endometrial cancer. Diagnosis of endometrial cancer usually requires an endometrial biopsy for a symptomatic patient [34]. Increased CA125 level was only found in about 60% of patients with endometrial cancer, often indicating extrauterine tumor spread, tumor grade, or lymph node metastasis [35].Kakimoto et al. revealed co-expressed CA125 and mesothelin in endometrial tumor tissue [36]. The co-expressed CA125 and mesothelin were most often observed in the histologic types of serous carcinoma, clear cell carcinoma, and carcinosarcoma. This co-expression was also associated with a poor prognosis. However, there was still insufficient evidence supporting CA125 as an independent prognostic marker of endometrial cancer [34].

For cervical cancer, increase of CA125 level was found more often in adenocarcinoma rather than squamous cell carcinoma, especially in mucinous adenocarcinoma [37]. Overexpression of MUC16 with MUC1 was reported as a poor prognostic factor [38]. However, only limited studies focused on the relation between CA125 and cervical cancer.

#### 4.1.2. Cancer Antigen 15-3 (CA15-3)

CA15-3 is the soluble form of MUC1 and is a highly glycosylated protein. It is a heterodimeric transmembrane protein commonly located on the apical of epithelium of both normal tissues and cancerous tissue. MUC1 glycoprotein is a complex molecule composed of a protein core and a large variable number of tandem repeats, which is the extracellular domain of MUC1 and high *O*-glycosylated [39]. The important distinguishing features between healthy and malignant cells are the differing chain lengths and structures of extracellular *O*-glycans [40].

CA15-3 has been a widely used diagnostic and prognostic biomarker of breast cancer since 1980s. An increased CA15-3 level is also observed in other adenocarcinomas, including ovarian, endometrial, pancreatic, gastric, and lung [39]. Moreover, increased CA15-3 level is also found in some benign conditions, such as type 2 diabetes mellitus or nephrotic syndrome.

Studies have found that tumor-associated CA15-3 played an important role in ovarian cancer metastasis and progression via many mechanisms [41]. The overexpression of CA15-3 on the surface of cancerous cells could disrupt cell—cell adhesion and function as an anti-adhesion molecule to induce cancer cells to escape from the tumor nest and form micrometastasis [42,43]. CA15-3 also plays a crucial role in tumor anti-apoptosis by increasing anti-apoptotic BclXL and PI3K/Akt pathways, as well as suppressing the release of apoptogenic factors from mitochondria [44]. It is also involved in epithelial—mesenchymal transition, an important mechanism for cancer invasion and metastasis [40]. For endometrial cancer, Engel et al. found that MUC1 increased cellular signaling through the epidermal growth factor receptor [45]. Therefore, CA15-3 could be a biomarker for ovarian and endometrial cancer, especially for tumor progression or metastasis, but more studies are needed to determine its fitness for clinical use. The sensitivity and specificity of CA15-3 alone for diagnosing ovarian cancer were low (sensitivity: 0.38, specificity: 0.25). Skates et al. combined four biomarkers (CA125, CA15-3, CA72-4, and Macrophage Colony-Stimulating Factor) together and reported good sensitivity and specificity for diagnosing early stageovarian cancer. CA15-3 may not be a sole screening marker for gynecologic cancer, but could be one choice of diagnostic marker [46].

CA15-3 is prevalently expressed in breast cancer. As a result, it was also a promising treatment target. The Mannan—MUC1 fusion protein-mediated stimulation of dendritic cells has been proven to be efficacious in phase I clinical trials [47]. However, in gynecologic cancer, the expression and importance of CA15-3 were still not well-established, so it may not be a candidate of treatment currently.

#### 4.1.3. Mucin 20 (MUC20)

MUC20 is transmembrane mucin that has been shown to regulate cell growth, differentiation, metastasis, adhesion, and invasive immune surveillance. MUC20 has been reported to correlate with the progression of various cancers [20,48,49], and is an independent prognostic factor for the poor survival rates of ovarian and endometrial cancer [20,48,50]. Overexpressed MUC20 activates the EGFR-STAT3 pathway, promotes epidermal growth factor receptor (EGF) expression, and induces a malignant phenotype of endometrial cancer [48]. In ovarian cancer, overexpressed MUC20 activates integrin β1 signaling and promotes FAK phosphorylation [20].

#### 4.1.4. Alpha-Fetoprotein (AFP)

AFP is an *N*-linked glycoprotein present in fetal serum. Generally, normal levels of AFP are below 10 ng/ml [51]. Elevated AFP levels could be found in different cancers, and hepatocellular carcinoma is the most common. Among gynecologic cancer, AFP was not a reliable tumor marker for screening or diagnosis, so it was usually combined with other markers to achieve better sensitivity and specificity [52]. Several ovarian germ cell tumors could produce AFP, including yolk sac tumor, embryonal carcinoma, and immature teratoma [35]. AFP is a reliable biomarker to distinguish malignant germ cell tumors from benign ovarian lesions. Similar to CA125, AFP could be applied as a biomarker for treatment effect and recurrence of germ cell tumors [53].

#### 4.1.5. Carcinoembryonic Antigen (CEA)

CEA is a 180 kDa GPI cell-surface-anchored glycoprotein normally expressed during fetal development. CEA dramatically decreases before birth, but has an average serum concentration below 2.5 ng/mL in adults [23]. Many conditions could cause elevation of CEA level, including smoking, thyroiditis, hypothyroidism, inflammatory bowel disease, liver cirrhosis, biliary obstruction, pancreatitis, renal failure, chronic obstructive lung disease, pneumonia, tuberculosis, and ovarian cyst [54]. Meanwhile, increased CEA could also be found in different malignancies, like colorectal, breast, ovarian, or lung cancer [55]. The sensitivity of CEA alone was insufficient for diagnosing ovarian cancer (at only 51.64%), although it had good specificity (94.0%) [56]. Studies in China recruited approximately 100 to 200 patients showed CEA combined other markers, e.g., CA-125, HE4, could improve diagnostic performance [52,56].

Some studies suggest the combination of CEA and CA125 for the evaluation of adnexal mass. Higher CEA levels may indicate the metastatic adnexal mass or mucinous histologic type [57]. However, it could not be a useful marker for differentiating borderline tumors from malignancies [58].

In cervical cancer, CEA is associated with advanced stages. Increased CEA levels indicated lymph node metastasis, especially in cervical adenocarcinoma. Therefore, CEA could be used as a prognostic marker in cervical cancer [55].

#### 4.1.6. Human Epididymis Protein 4 (HE4)

HE4, a small secretory *N*-linked glycoprotein (about 25 kDa), encoded by the WFDC2 gene, is another ovarian cancer marker intensely studied in the last several years and was recently introduced in clinical use [24]. In 2008, the FDA approved HE4 for monitoring patients with an established diagnosis of ovarian cancer, but not for screening early stage ovarian cancer in asymptomatic women [59]. Some studies revealed a better diagnostic performance of ovarian cancer than CA125, especially for detecting recurrence of cancer [60]. One meta-analysis revealed an overall sensitivity of 79% and a specificity of 93% for HE4, and an overall sensitivity of 79% and specificity of 78% for CA125 [30].

The level of HE4 could be measured by a manual enzyme immunometric assay (EIA) or other immunological methods, such as electrochemiluminescent (ECLIA) or chemiluminescent microparticle immunoassay (CMIA) [61,62]. The cutoff level of 70 pmol/L is often used for premenopausal patients, and 140 pmol/L is for menopausal patients. Nowadays, using HE4 and CA125 as dual markers could improve performance for distinguishing malignancy ovarian tumors from borderline or benign lesions [63,64]. The Risk of Ovarian Malignancy Algorithm [58] test is one diagnostic tool that combines CA125, HE4, and menopausal status with a sensitivity of 93.8% and specificity of 74.9% [65]. The FDA approved the ROMA test in 2011 for predicting the risk of ovarian malignancy in women presenting with pelvic masses. However, the biomarker still has limitations in the UK for pre-operative evaluation of ovarian tumor and for distinguishing the malignant tumor from the benign and borderline tumor [66].

Several studies also investigated HE4 as a biomarker of endometrial cancer [67]. The meta-analysis showed good specificity, acceptable accuracy, but limited sensitivity (sensitivity, specificity, and AUC of 65%, 91% and 0.84, respectively). Due to limited sensitivity, HE4 is not proved as a diagnostic marker for endometrial cancer. Studies also found that some poor prognostic factors of endometrial cancer, including stage, lymph node metastasis, and the presence of a lymph-vascular space invasion were associated with an increase of HE4 level [67]. Meta-analysis also unveiled the strong association between HE4 level and survival of endometrial cancer patients [68]. As a result, some studies suggested HE4 could be a promising prognostic marker for endometrial cancer.

#### 4.1.7. Squamous Cell Carcinoma Antigen (SCC-Ag)

SCC-Ag is a member of the endogenous serine protease inhibitor (SERPIN) family and was first isolated by Kato and Torigoe in the 1970s from squamous cell carcinoma tissues of the uterine cervix [69]. It is a 48 kDa cytoplasmic glycoprotein found in normal squamous epithelia and is a mixture of two isoforms. In conjunction with clinical evaluation, SCC-Ag assessment may serve as a non-specific tumor marker for detecting and monitoring diverse squamous cell carcinomas, including those originating in the head and neck, esophagus, cervix, and lung [9].

Recent studies also showed that SCC-Ag played an important role in inflammatory diseases [70]. The low sensitivity and specificity of SCC-Ag, particularly for early stage cervical cancer, decrease its use for screening or diagnosis. However, regarding the prognosis, SCC-Ag is an important prognostic factor for cervical cancer, and its high expression is significantly correlated with a poor disease prognosis [71].

#### 4.1.8. Glypican-3 (GPC3)

Glypicans (GPCs) are a family of heparan sulfate proteoglycans attached to the cell membrane via a GPI anchor. Glypicans interact with multiple ligands, including morphogens, growth factors, chemokines, ligands, receptors, and components of the extracellular matrix through their heparan sulfate chains and core protein [72]. Humans have six glypican family members, referred to as glypican-1 (GPC1) through glypican-6 (GPC6). GPC1 is highly expressed in solid tumors, especially squamous cell carcinomas [73], and is thought to be associated with disease progression. Like AFP, GPC3 is an oncofetal protein expressed only in the placenta and fetal tissues. Thus, it was proposed to be a promising tumor marker for the diagnosis of hepatocellular carcinoma [74]. GPC3 is also a biomarker for ovarian germ cell tumors, specifically yolk sac tumors and choriocarcinoma [75]. Other studies have shown that GPC3 expression was strongly associated with clear cell ovarian carcinomas, and high GPC3-expression was significantly associated with unfavorable outcomes in cases with loss of ARID1A [76,77].

#### 4.1.9. Tissue Factor (TF)

Tissue factor (TF, coagulation factor III, CD142) is a single-stranded and intact plasma membrane glycoprotein of 43–45 kDa in the cytokine receptor superfamily, which is the primary physiological initiator of normal blood coagulation. TF exhibits a different nonuniform tissue distribution and is aberrantly expressed in solid cancers. TF is thought to contribute to disease progression through its procoagulant activity and its capacity to induce intracellular signaling in complex with factor VIIa [78]. TF is highly expressed in cervical cancer tissues, and high expression of TF may enhance the invasion and metastasis of cervical cancer cells [79,80].

Tisotumab vedotin, an antibody-drug conjugate, targets TF by delivering the cytotoxic agent monomethyl auristatin E directly into tumor cells [78]. The FDA has approved tisotumab vedotin for treating adult female patients with recurrent or metastatic cervical cancer with disease progression on or after chemotherapy.

### 4.2. Glycan Markers

In the endoplasmic reticulum and Golgi apparatus, glycosyltransferases and glycosidases form glycoconjugates. The synthesis of glycoconjugates is a dynamic process controlled by substrate availability, enzyme activity, levels of gene transcription, and enzyme location within the organelles [7,8]. With the catalyst of polypeptide-GalNAc transferases (pp-GalNAcTs, GALNTs), a single GalNAc residue is transferred to Ser/Thr residues of specific proteins, forming the Tn antigen, which finally forms four core structures referred to as Core 1 through Core 4 (Figure 3).

The structure of glycans can alter as cancer cells evolve. This phenomenon consists of incomplete synthesis, which refers to truncated glycosylation that produces short *O*-glycans such as T, Tn and Sialyl-Tn, and neo-synthesis, which produces abnormal glycosylation patterns such as sialyl Lewis X (sLe^X^) and sialyl Lewis A (sLe^A^) [11,81].

#### Cancer Antigen 19-9 (CA19-9)

CA19-9 contains a tetrasaccharide carbohydrate known as sLe^A^ [82]. sLe^A^ is synthesized by glycosyltransferases which sequentially bind the monosaccharide precursors onto both *N*-linked and *O*-linked glycans [83]. CA19-9 is a large glycoprotein secreted by the pancreas and bile ducts, gastric system, colon, and endometrial epithelium.

The normal physiological level of serum CA19-9 is under 37 U/mL. Serum CA19-9 levels are elevated in 70% to 80% of pancreatic cancer patients. They may be more suitable for the diagnosis of pancreatic cancer rather than for predicting prognoses of patients with an established diagnosis [84]. The FDA approved it to be a marker for pancreatic ductal adenocarcinoma in 2002 [29]. Increased CA19-9 level was also found in ovarian cancer, especially for the mucinous histologic type, with a sensitivity and specificity of 52.7% and 83.8%, respectively [58]. However, elevation of CA19-9 level could also be found in benign or borderline ovarian tumors, so this marker could not distinguish benign, borderline tumors from malignancy [85].

For diagnosing endometrial cancer, CA19-9 had only limited sensitivity (16.3%) [86]. Only a few studies also investigated CA19-9 as a biomarker for endometrial cancer. Due to its limited performance in gynecologic cancer, CA19-9 remains the only pancreatic cancer marker in clinical use.

### 4.3. Enzyme Markers

Although the enzyme markers do not belong to glycoprotein, they play important roles in glycosylation within cancer cells. Therefore, we still briefly introduced some enzyme markers in this review.

The glycoproteins are glycosylated by the activity of glycosyltransferase. Sometimes the occurrence of altered glycosylation is a change in the expression of glycosyltransferases. Thus, although most of the clinically useful cancer biomarkers are glycoproteins, changes in glycosylation machinery, such as glycosyltransferases, could be used as cancer biomarkers.

*N*-glycosylation occurs at the Asn-X-Ser/Thr sites of proteins, which are sequentially modified via the action of α-mannosidases, *N*-acetyl-glucosaminyltransferases (Mgat) I, II, IV, and V (encoded by Mgat1, 2, 4, and 5, respectively) that each add GlcNAc from uridine diphosphate *N*-acetylglucosamine (UDP-GlcNAc) to a specific mannose in the core of a *N*-glycan. Niimi et al. [87] reported that Mgat4a protein expression was strong in invasive mole and choriocarcinoma and Nishino et al. [88] confirmed that Mgat4 increased invasion of choriocarcinoma.

Mucin-type *O*-glycans are modified with a GalNAc sugar at the hydroxyl group of Ser/Thr residues of proteins by a large family of GALNTs. GALNTs consist of at least 20 members in humans, namely GALNT1 to 20 [89]. Studies have shown that mislocalization and dysregulation of GALNTs expression result in aberrant glycosylation in cancer cells, denoting the critical roles of GALNTs in regulating cancer’s behavior [90,91,92]. GALNTs have differential but partly overlapping substrate specificities, which suggest different disease progressions catalyzed by distinct GALNTs in a cell- or tissue-specific way [93]. For instance, GALNT6 was upregulated in breast cancer, and might contribute to mammary carcinogenesis through aberrant glycosylation and stabilization of MUC1 [94]. GALNT14 was overexpressed in colorectal carcinoma and pancreatic cancer. It was associated with altered sensitivity to TRAIL-induced apoptosis through modulation of the *O*-glycosylation of death receptors on these tumor cells [95]. In gynecologic cancers, a growing number of studies have reported that abnormal mucin-type *O*-glycosylation mediated by GALNTs can promote cancer cell proliferation, survival, and metastasis [96,97,98,99].

#### 4.3.1. *N*-acetylgalactosaminyltransferase 6 (GALNT6)

GALNT6 has been extensively studied for its implication in the malignant transformation and metastasis of epithelial cancers, especially in breast cancer, where it has been suggested as a novel marker for detection and a potential therapeutic target [100,101].

In ovarian cancer, higher GALNT6 levels are significantly associated with poorer patient survival rates. Lin et al. found that GALNT6 expression is associated with a poor prognosis of ovarian cancer and enhances the aggressive behavior of OVCA cells by regulating EGFR activity [98]. Sheta et al. also showed that altered expression of GALNT6 is associated with disease progression and poor prognosis in women with high-grade serous ovarian cancer [99]. In endometrial cancer, GALNT6 is an indicator of good prognosis and noninvasive tumors in patients with endometrial carcinoma [102].

#### 4.3.2. Core 1 β1, 3 Galactosyltransferase 1 (C1GALT1)

C1GALT1 is a primary glycosyltransferase needed in the biosynthesis of mucin-type *O*-glycosylation. C1GALT1 transfers galactose to Tn-antigen (GalNAca-Ser/Thr) to form T-antigen (core 1 structure). T-antigen is further modified by adding other sugars via other glycosyltransferases to create the complex mucin-type *O*-glycans. Dysregulation of C1GALT1 is involved in cancer development and progression by promoting either *O*-glycan truncation or elongation. C1GALT1 overexpression is associated with tumor growth, metastasis, and poor prognosis, and has been observed in various cancers such as ovarian [103], hepatocellular [103], breast [104], colorectal [103], and gastric cancer [105].

## 5. Summary

Gynecologic cancers are still terrifying for women because of poor prognoses and high mortality rates of advanced stages. Early detection of disease with non-invasive methods is still a challenge for clinicians. Glycosylated biomarkers, expressed aberrantly in cancer cells, are a promising tool for cancer detection and prognosis determination. Many biomarkers had already been well-researched for their value and roles in cancers. For gynecologic cancers, a small handful of tumor-associated antigens, such as CA125, CA19-9, HE4, and SCC, have been routinely used as tumor markers. The use of these biomarkers for the diagnosis and monitoring of gynecologic cancers has significant advantages, as it is a quantitative, objective, and comparable measure that is also convenient and relatively inexpensive compared with histology and imaging. However, none of the single serum marker has sufficient sensitivity for screening gynecologic cancers. The use of combination marker panels is frequently selected in the clinic due to the high sensitivity and specificity, but a combination panel has increased costs. Some new biomarkers being investigated currently, such as MUC20 and GALNT6, seem to be clinically useful. The major disadvantages are that these markers require considerable expertise and more expensive instrumentation. Along with the recently developed instrumentation and platform, newer biomarkers could be found to fulfill the goal of early detection and prognosis prediction. The existing markers could also be a promising target of treatment. There is still much potential application of glycosylated markers in gynecologic cancers.

## Figures and Tables

**Figure 1 diagnostics-12-03177-f001:**
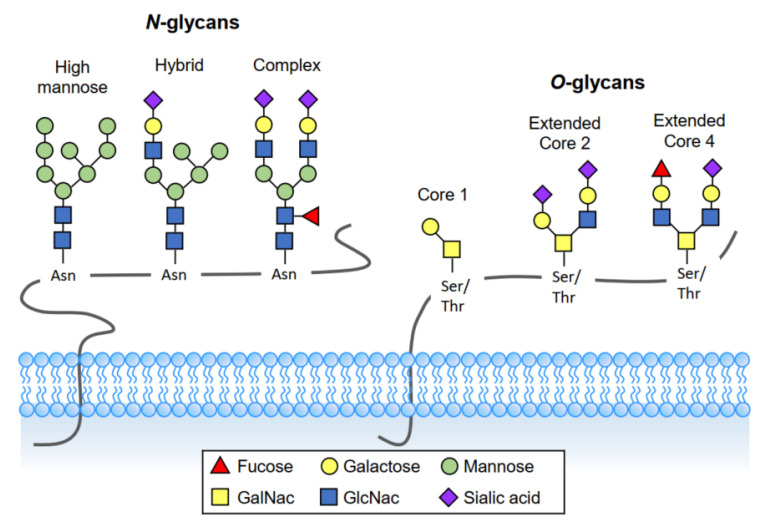
Two major types of protein glycosylation. *N*-glycans consist of N-acetylglucosamine (GlcNAc) attached by a β1-glycosidic linkage to the amino group of asparagine (Asn). Mucin-type *O*-glycosylation, consisting of *O*-glycans attachment via *O*-linked Nacetylgalactosamine (GalNAc) to the hydroxyl oxygen of serine/threonine (Ser/Thr) residues.

**Figure 2 diagnostics-12-03177-f002:**
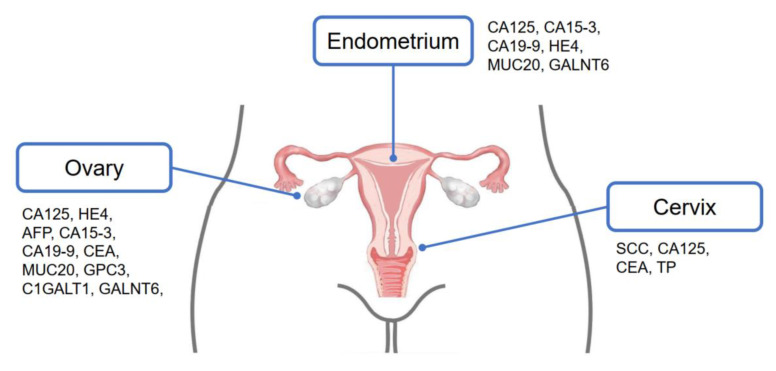
FDA-approved and potential new biomarkers of gynecologic cancers.

**Figure 3 diagnostics-12-03177-f003:**
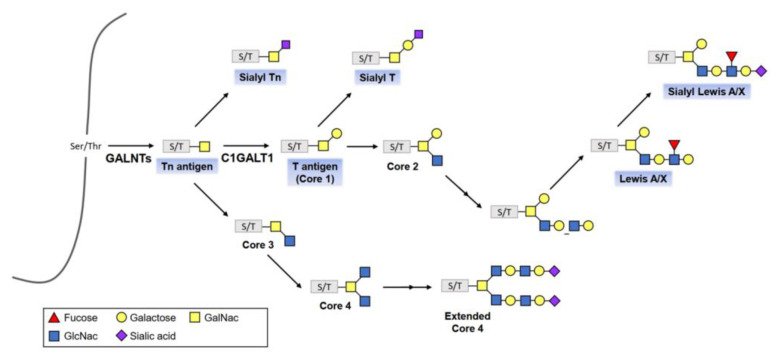
The *O*-linked glycosylation biosynthetic pathway. Cancer-associated structures are highlighted with blue boxes. With the catalyst of polypeptide-GalNAc transferases (GALNTs), a single *N*-acetylglucosamine (GlcNAc) residue is transferred to serine/threonine (Ser/Thr) residues of specific proteins, forming the Tn antigen, which finally forms four core structures referred to as Core 1 through Core 4.

## Data Availability

Not applicable.

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
