# Peer review of "Protein Glycosylation as Biomarkers in Gynecologic Cancers"

_diagnostics, 2022, doi:10.3390/diagnostics12123177_

Round 1
Reviewer 1 Report
This manuscript describes a review of biomarkers for gynecologic cancer. The contents are clear and easy for general readers to understand. However, biomarkers described in this manuscript are not always glycoprotein and not refer its glycan structure. A reviewer feels additional description improves this review manuscript.
In terms of general biomarkers such as CA125, CA53, CEA and AFP, the authors should show their specificity and sensitivity for diagnosis of gynecologic cancer according to previous reports using a big cohort.
In terms of glycan biomarkers, Mgat4 and chorionic carcinoma, Mgat5 and cancer metastasis of gynecologic cancer, fructosyltransferase and ovarian cancer should be included. These papers are well known in glycobiology research.
Reviewer 2 Report
This is a review regarding the role of glycosylated markers in gynaecological cancers. The methodology of literature search is not clear. I have the following comments:
1. What search strategies were used for this review?
2. Which databases were searched?
3. A systematic review should be performed when submitting manuscripts to a journal like this.
4. It would be more prudent to divide this text into three parts (ovarian, endometrial and cervical) and discuss a possible role of these biomarkers in these cancers.
5. In the introduction section the authors should discuss what the existing screening strategies are for these cancers. CA125 has been proven to be ineffective in ovarian cancer screening and respective studies should be mentioned with regards to this topic.
6. The summary should be rewritten. The authors should provide an explanation regarding the limitations of these markers in diagnosing gynaecological cancers. Is it cost-beneficial to use these markers in screening strategies and in follow-up?
Round 2
Reviewer 2 Report
All comments have been addressed and I
believe the manuscript is now suitable for
publication
Author Response
Thank you for your kind and helpful advise.